# Learning Strategic Network Emergence Games

**Rakshit Trivedi**
Georgia Institute of Technology
rstrivedi@gatech.edu

**Hongyuan Zha**
AIRS and Chinese University of Hong Kong, Shenzhen
zha@cc.gatech.edu

## Abstract

Real-world networks, especially the ones that emerge due to actions of animate agents (e.g. humans, animals), are the result of underlying strategic mechanisms aimed at maximizing individual or collective benefits. Learning approaches built to capture these strategic insights would gain interpretability and flexibility benefits that are required to generalize beyond observations. To this end, we consider a game-theoretic formalism of network emergence that accounts for the underlying strategic mechanisms and take it to the observed data. We propose MINE (**M**ulti-agent **I**nverse models of **N**etwork **E**mergence mechanism), a new learning framework that solves Markov-Perfect network emergence games using multi-agent inverse reinforcement learning. MINE jointly discovers agents' strategy profiles in the form of network emergence policy and the latent payoff mechanism in the form of learned reward function. In the experiments, we demonstrate that MINE learns versatile payoff mechanisms that: highly correlates with the ground truth for a synthetic case; can be used to analyze the observed network structure; and enable effective transfer in specific settings. Further, we show that the network emergence game as a learned model supports meaningful strategic predictions, thereby signifying its applicability to a variety of network analysis tasks.

## 1 Introduction

Machine Learning methods for network analysis [1, 18, 47, 54] typically operate on the stochastic assumption about the nature of underlying mechanisms that govern the emergence of observed networks. Several networks, in spite of their different origins, indicate large commonalities among their structural properties (e.g. diameter, clustering coefficient, etc.). Solution approaches to learn from network data, therefore, rely on optimizing the likelihood of observing specific structural properties and often use surrogate objectives parameterized by modules that capture these properties, achieving notable success across different areas [6, 7, 9, 23, 27, 39]. On the other hand, many real-world networks, that emerge due to actions of animate agents, are the result of strategic behavior of individuals rather than based on probabilities. For instance, economic partnerships between financial organizations, social collaborations at work and trade between countries, all pertain to strategic actions adopted by individual entities. The network emerging out of such relationships may not correspond to common structural parameters. Hence, it would be beneficial to model the learning problem as an equilibrium, an under-explored area within the network learning literature.

Network Emergence (Creation) Games [11] provide a formal interpretable framework to characterize and analyze the decentralized process among many interacting agents, whose outcome is the observed network. This process assumes that agents follow strategic behavior in forming links with other agents and the interacting agents are considered to be inherently selfish with some capacity allowing for emergent local coordination between neighboring agents. While serving as an elegant theoretical tool to explain network emergence process, the practical applicability of Network Emergence Games beyond building stylized simulators is often hindered due to several limitations:

Existing game theoretic approaches to analyze networks [11, 38, 20, 34, 24] do not directly learn from the observed network data, instead their model is hand-designed with specifications based on assumptions and intuitions about the real observations. Such specifications manifest in the form of an agent-specific *payoff (utility)*, wherein an agent is assumed to optimize this utility so as to maximize their individual benefits based on their position in the network. Simple forms of payoff function only capture a subset of properties related to the strategic behavior of agents, but modeling complex properties using such hand-designed utility functions is often prone to misspecification. Further, most network games assume access to the entire network (complete information) whereas in real world networks, agents often deal with incomplete information mainly limited to its neighbors. Additionally, the emergence procedure in such games is not considered to be sequential (i.e. all agents announce all their links in one shot and the resulting network is analyzed for equilibrium after which the game is restarted). To our knowledge, while the problem of incomplete information in network games is less explored, recent works [24] have proposed Markov version of network games that consider network emergence as a sequential process and we discuss them in detail later.

**Our Approach.** In this work, we seek to combine the interpretability merits of game theoretic modeling with practical usefulness of data-driven learning and propose an algorithmic framework, MINE, to learn strategic network emergence games from the observed network data. Concretely, our data consists of a graph structure between $n$-players (represented by the nodes), where the observed structure is the result of strategic interactions between these $n$ players under some unknown payoff. A key novelty of MINE is to explicitly incorporate the network game dynamics into the learning framework tasked to jointly discover both the agents' strategies and unknown payoff mechanism that may have governed the emergence of the observed network. Learning the payoff directly from the observed data (inverse problem) allows to model complex mechanisms – often missed by the hand-designed specifications or difficult to express in an explicit form in the first place. We consider this network emergence game being played in a Markov setting where the agents interact with each other in a sequential manner to achieve Markov Perfect equilibrium networks and the strategies of agents at any step only depends on the current state of the network. To develop a practical learning framework, we leverage on Markov Quantal Response equilibrium (MQRE) as a solution concept and base our algorithm building on recently proposed inverse reinforcement learning technique [50] for solving Markov games, where the learned reward is interpreted as the payoff mechanism. Solving for equilibrium in our setting can then be viewed as searching for agents' strategy profiles (policies) that leads to equilibrium network (under the learned reward) from the set of all feasible networks which is a combinatorial search task. For effective learning in graph structured environment, we use graph neural network (GNN) [41] based state representation, continuous action space and maximum entropy RL [16] algorithm to update policies in the inner loop of reward learning.

A key outcome of our data-driven learning of network emergence games is its ability to facilitate *interpretability* (the discovered reward function can be analyzed to characterize the observed network properties) and *generalization* (the learned modules can be used beyond observed set of agents in specific settings) – both of which are hallmarks of human intelligence and highly desirable of any learning framework. Additionally, as MINE focuses on strategic mechanisms, it supports downstream strategic prediction tasks, signifying its practical usefulness for network analysis of real-world agent-based networks. In the experiments, we focus on the above properties of MINE and seek to address the following questions: (i) Can MINE effectively discover the payoff mechanisms useful to analyze the characteristics of observed network?; (ii) How well does the learned payoff facilitate generalization across different set or number of players? and (iii) Can the learned network emergence game serve as an effective predictive model? We answer these questions in the positive by analyzing reward to characterize a strategic network structure, performing in-domain transfer for trade and movie networks and evaluating strategic link prediction performance across different networks.

## 2 Preliminaries

### 2.1 Markov Network Emergence Game

We consider an $n$-player Markov Network Formation Game, where agents form links with each other to maximize their individual payoffs. The game is played in a sequential manner, where at each step, the agents announce the links they want to form and their current strategies are dependent only on the current state of the network - the Markov property. A general $n$-player Markov game [26] is defined as $(\mathcal{S}, \{\mathcal{A}_i\}_{i=1}^n, \{r_i\}_{i=1}^n, \mathcal{P}_T, \nu, \gamma)$ where $\mathcal{S}$ is the state space and $\mathcal{A}_i$ is the action space for

agent $i$. The function $\mathcal{P}_T : \mathcal{S} \times \mathcal{A}_1 \times ... \times \mathcal{A}_n \to \mathcal{P}_T(\mathcal{S})$ specifies the transition process between states, where $\mathcal{P}_T(\mathcal{S})$ denotes probability distribution over set $\mathcal{S}$. $\nu \in \mathcal{P}_T$ specifies an initial state distribution and $\gamma$ is the discount factor. A state $s_t$ of the game at time $t$ transitions to state $s_{t+1}$ with probability $\mathcal{P}_T(s_{t+1}|s_t, a_1, ..., a_n)$ due to agents' actions $\{a_1, ..., a_n\}$. Each agent $i$ receives a reward given by the function $r_i : \mathcal{S} \times \mathcal{A}_1 \times ... \times \mathcal{A}_n$. We use bars to indicate joint quantities over all agents – $\bar{\pi}$ denote the joint policy, $\bar{r}$ denote rewards of all agents and $\bar{a}$ denote actions of all agents. Further, subscript $-i$ denotes all agents other than $i$ – the tuple $(a_i, \bar{a}_{-i})$ denote actions of all agents. Each agent $i$ aims to maximize its individual payoff $u_i$, instantiated by the expected sum of discounted rewards, $u_i = \mathbb{E}_\pi \left[ \sum_{t=1}^T \gamma^t r_{i,t} \right]$, where $r_{i,t}$ is the reward received $t$ steps into the future by agent $i$. Each agent selects actions according to its stochastic policy $\pi_i : \mathcal{S} \to P(\mathcal{A}_i)$, where $P(\mathcal{A}_i)$ is the distribution over agent $i$'s actions space. Further, for each agent $i$, the expected return for a state-action pair is defined as:

$$u_i^{\pi_i, \bar{\pi}_{-i}}(s_t, \bar{a}_t) = \mathbb{E}_{s_{t+1:T}, \bar{a}_{t+1:T}} \left[ \sum_{l \geq t} \gamma^{l-1} r_i(s_l, \bar{a}_l)|s_t, \bar{a}_t, \bar{\pi} \right]. \tag{1}$$

**Network Emergence Game.** We consider a partially observable Markov Game, where each agent has access only to its local observations. For a set of $n$ interacting agents, let $\mathcal{G} \subset \{0,1\}^{n \times (n-1)}$ denote the set of all feasible networks. Let $G = (\mathcal{V}, \mathbf{A} \in \mathcal{G}, \mathcal{X})$ specify one such feasible network, where $|\mathcal{V}| = n$ is the set of agents (vertices) in the game, $\mathbf{A}$ is the adjacency matrix specifying the link structure between agents and $\mathcal{X}$ denote the feature set for agents. These features may model the innate characteristics (basic profile) of agents that contribute to their strategic behavior. Below we outline the decision process that incorporates the game-theoretic network emergence dynamics:

**State** $\mathcal{S}$: The state of the game $s_t$ at any time $t$ is the graph structure $G_t = (\mathcal{V}, \mathbf{A}_t, \mathcal{X})$, where $\mathbf{A}_t$ contains information of the graph structure at time $t$. $G_0 = (\mathcal{V}, \mathbf{A}_0 = \mathbf{0}, \mathcal{X})$ defines the initial state of the game $s_0$. Any agent $i$ can only access its local observation $o_{i,t} = (\mathcal{N}_{i,t}, \mathcal{X}_{\mathcal{N}_{i,t}}) \in \mathcal{O}$ from the game's overall state $s_t \in \mathcal{S}$, where $\mathcal{N}_{i,t} = \{j|\mathbf{A}_t^{ij} = 1\}$ is the neighborhood of agent $i$ at time $t$.

**Action** $\mathcal{A}$: A step $t$ in the game involves each agent announcing their intentions to form the links with other agents. We map this action to a continuous low-dimensional vector $\mathbf{a}_{i,t} \in \mathbb{R}^d$ where $d \ll n$ is the action dimension). The vectors announced by each agents are then matched externally (details discussed later) to compute the links that finally emerge out of that step. This action space is inspired from recent work on Geometric Network Creation Games [3].

**Transition Dynamics** $\mathcal{P}_T$: Let $\bar{e}$ denote set of joint edge operations that are derived from joint action profile $\bar{a}$ for all agents at state $s_t$. The transition function $\mathcal{P}_T$ is defined such that the $\bar{a}$ profile obtained for all agents at state $s_t = (\mathcal{V}, \mathbf{A}_t, \mathcal{X})$ produces the next state $s_{t+1} = (\mathcal{V}, \mathbf{A}_{t+1}, \mathcal{X})$. Here, $\mathbf{A}_{t+1} = \mathbf{A}_t \odot \bar{e}$ where $\odot$ represents all the edge operations (such as creation, maintenance or severance of an edge for the game) applied to adjacency $\mathbf{A}_t$ to modify the network structure.

**Reward** $r$: A key objective of our work is to infer agent's payoff from the observed network and hence we do not impose any strict functional form on the reward function, instead learn the reward function $r_i$ (for each agent $i$) directly from the data. The reward parameterization controls the shared properties of the utility function across agents. Further, we consider the localized utility setting where the reward $r_i(o_{i,t}, a_{i,t})$ for agent $i$ is computed only with respect to its current neighborhood.

## 2.2 Solution Concept for Network Emergence Games

Network emergence games focus on analyzing the construction of equilibrium networks where no agent want to locally change the network [38]. To setup our reinforcement learning (RL) procedure, the first step is to specify an appropriate equilibrium concept for characterizing the trajectories distribution induced by the reward function. We focus on Markov Perfect Equilibrium (MPE) [42, 28–30], as it directly relates to this work and discuss others in Appendix D. This concept has been investigated and formalized in recent works in stochastic game formulation for network emergence [24, 10]. MPE admits properties that map directly to reinforcement learning setting, as discussed below.

**Markov Perfect Equilibrium.** The definition of network emergence policy $\pi_i$ for each agent $i$ concretely specifies agents' Markov strategies – a Nash equilibrium (NE) in which is referred as Markov Perfect Equilibrium. While existence of an MPE has been long established for stochastic

games [12, 45, 43], a straightforward application of Bellman's optimality principle [2] shows that solving for MPE directly maps to recursive procedure of learning optimal joint policy $\bar{\pi}^*$ by optimizing individual reward $r_i$ using the RL procedure (details provided in Appendix A). However, solving for MPE requires solving for NE at each state, which is not amenable to learning due to discontinuous characteristics of NE with respect to payoff matrix. Further, NE assumes all agents to be perfectly rational which is often not the case for agents participating in real-world network emergence. Both these difficulties are addressed by Quantal Response Equilibrium (QRE) and its logistic version [31, 32], which is stochastic generalization of NE. Specifically, QRE accounts for bounded rationality using a parameter $\lambda$ and models payoff matrices injected with noise, thereby introducing smoothness useful for gradient based approaches [25]. For stochastic games, QRE has been extended to Markov version by [5, 10] and referred as Markov Quantal Response Equilibrium (MQRE).

**Logistic Markov Quantal Response Equilibrium.** The most interesting version of MQRE is its logistic version (LMQRE) which arise from the noise that is i.i.d according to Gumbel distribution with parameter $\lambda \in \mathbb{R}^+$, which also controls the rationality of agents. An MLQRE $\bar{\pi}^*$ can then be expressed in closed form as a solution to the following system of equations: For all states $s_t \in \mathcal{S}$, all agents $i$ and all actions $a \in \mathcal{A}_i$:

$$\bar{\pi}_i^*(a_i|s) = \frac{e^{\lambda \cdot \hat{u}_i(s,a,\bar{\pi}_{-i}^*)}}{\sum_{a' \in \mathcal{A}_i} e^{\lambda \cdot \hat{u}_i(s,a',\bar{\pi}_{-i}*)}} \; ; \; V_i^*(s) = \sum_{a' \in \mathcal{A}_i} \bar{\pi}_i^*(a_i|s) \cdot \hat{u}_i(s,a',\bar{\pi}_{-i}^*) \qquad (2)$$

where $\hat{u}_i$ represent the noise injected payoff version: $\hat{u}_i(s,a,\bar{\pi}) = u_i(s,a,\bar{\pi}) + \varepsilon_i(s,a)$ that an agent $i$ is assumed to perceive in a quantal response framework. Here, $u_i$ is the expected payoff from playing action $a$ for agent $i$ in state $s$, given strategies of other players and is expressed as : $u_i(s,a,\bar{\pi}) = r_i(s,a,\bar{\pi}) + \gamma \sum_{s' \in \mathcal{S}} P_{ss'}(a_i,\bar{\pi}_{-i}) \cdot V_i(s')$. $\lambda$ can be interpreted as the precision with which the agents perceive the payoffs. When $\lambda \to 0$, the equilibrium is fully noisy and the agents with select actions uniformly at random. When $\lambda \to \infty$, the agents will choose actions in best response manner (greedily). [10] has shown that the limit point of LMQRE converges to Markov perfect equilibrium. That is, $\bar{\pi}^{**} = \lim_{\lambda \to \infty} \bar{\pi}^*(\lambda)$ is a Markov perfect equilibrium for some LMQRE $\bar{\pi}^*(\lambda)$. This establishes LMQRE as an appropriate equilibrium concept to use in the RL setting for solving Markov Perfect Network emergence game which is our case. We discuss more details on MQRE, its convergence to Markov Perfect equilibrium and properties of $\pi_i$ in the Appendix A.

## 2.3 Multi-Agent Inverse Reinforcement Learning

Most game-theoretic methods hand-design a reward function for theoretical analysis, but for real-world networks, it is often difficult to specify the explicit form of such reward mechanism. One of the key contributions of this paper is to discover this reward function from the observed networks. Specifically, our goal is to jointly learn the strategy profile for agents that leads to the emergence of observed network and discover the latent payoff function. To achieve this, we leverage maximum entropy (MaxEnt) inverse reinforcement learning (IRL) [55] which aims to learn a reward function that rationalizes the expert behaviors with least commitment. Let $\mathcal{D}$ denote expert demonstrations provided by $n$ experts in the $n$-player Markov game. $\mathcal{D}$ is realized as a set of $M$ trajectories $\{\tau_j\}_{j=1}^M$, where $\tau_j = \left\{(s_j^t, \bar{a}_j^t)\right\}_{t=1}^T$ denote an expert trajectory collected by sampling $s^1 \sim \nu(s), \bar{a}^t \sim \pi_E(\bar{a}^t|s^t), s^{t+1} \sim P(s^{t+1}|s^t, \bar{a}^t)$. $\mathcal{D}$, obtained from observed graph for network emergence game, contains all the available supervision for the learning procedure. Denoting MaxEnt IRL function as IRL$(\pi_E)$, we have: IRL$(\pi_E) = arg\max_{r \in \mathbb{R}^{S \times A}} \mathbb{E}_{\pi_E}[r(s,a)] - RL(r)$, where $RL(r) = \max_{\pi \in \Pi} \mathcal{H}(\pi) + \mathbb{E}_\pi[r(s,a)]$ and $\mathcal{H}(\pi) = \mathbb{E}_\pi[-\log \pi(a|s)]$ is the policy entropy. The forward RL problem in the inner loop makes the above procedure less efficient and hence various improvements have been proposed in the literature [13, 14]. For this work, we consider MA-AIRL [50], a recently proposed IRL algorithm for multi-agent setting. MA-AIRL uses Logistic Stochastic Best Response Equilibrium (LSBRE) as a solution concept to characterize the trajectory distributions induced by the reward functions of agents $\{r_i(s,\bar{a})\}_{i=1}^N$. Given a Markov game with horizon $T$, LSBRE is defined as a sequence of $T$ stochastic policies $\{\pi^t\}_{t=1}^T$, where each joint policy

$\bar{\pi} : \mathcal{S} \to P(\mathcal{A}_1 \times ... \times \mathcal{A}_n)$ is given by: $\bar{\pi}^t(a_1, ..., a_n | s^t) = P\left(\bigcap_i \{z_i^{t,(\infty)}(s^t) = a_i\}\right)$. Here, $z_i$ is a mapping from state to action:

$$z_i^{t,(k+1)}(s^t) \sim P_i^t(a_i^t | \bar{a}_{-i}^t = \bar{z}_{-i}^{t,(k)}(s^t), s^t) = \frac{\exp\left(\lambda Q_i^{\bar{\pi}^{t+1:T}}(s^t, a_i^t, \bar{z}_{-i}^{t,(k)}(s^t))\right)}{\sum_{a_i'} \exp\left(\lambda Q_i^{\bar{\pi}^{t+1:T}}(s^t, a_i'^t, \bar{z}_{-i}^{t,(k)}(s^t))\right)} \quad (3)$$

where $\lambda \in \mathbb{R}^+$ is noise parameter controlling rationality of agents, $k$ is the step and $\{P_i^t\}_{i=1}^N$ specifies set of conditional distributions. We specifically note the close relation of the form of $z_i$ in LSBRE with that of $\pi_i$ in LMQRE that allows us to design our practical algorithm building on MA-AIRL [50] while using LMQRE as a solution concept for network emergence games. We discuss more details on connection between LMQRE and LSBRE in the Appendix A.

## 3 Proposed Model

In this section, we first describe the architecture details of the MINE model and then outline the training procedure that jointly learns both the reward function and the network emergence policy.

### 3.1 Architecture

For a network emergence game, the performance of agents depend on effectively learning over the graph structure of the problem. To this end, we design a graph neural network [41] based policy network that embeds the observation into a continuous vector, further processed by the next layers of policy network to output the action vector. We outline the details on our structured strategy network below and specify the details on environment updates based on the continuous action vector. We also outline the parameterization of the reward function and its connection to game-theoretic approaches.

*Structured Strategy Network.* We leverage structured policy network [48] to design and implement a mapping function from state $s_t = G_t$ to the embedding matrix $\mathbf{H} \in \mathbb{R}^{n \times d}$ where each row of the matrix represent the embedding $\mathbf{h}_i$ for agent $i$. The mapping starts from the initial agent features $\mathbf{h}_i^{(0)}$ which are problem dependent. A $P$-step message passing procedure updates these features by aggregating information from $P$-hop neighborhood of agent $i$ as follows: $\mathbf{h}_i^{(p+1)} = U\left(\mathbf{h}_i^{(p)}, A\{m_{ij}\}_{j \in \mathcal{N}(i)}\right)$, where $U$ is the update function (GRU or MLP), $A$ is an aggregate function (sum, mean or max pooling), $m_{ij} = \mathbf{h}_j^{(p)}$ is the message on the edge between agents $i$ and agent $j$ and $\mathcal{N}(i)$ is the neighborhood agent set for agent $i$. The update to all agents' embeddings occur at the end of the environment step and the policy function takes the agent $i$- specific local observations (now mapped to embeddings) as input to compute the action for agent $i$. For computing a graph/subgraph embedding, we use an attention based aggregation of the participant agents' embeddings from last message passing iteration $\mathbf{h}_i^{(P)}$. Next, we design a stochastic policy that takes as input observation $o_{i,t}$ and outputs a link formation action $a_{i,t}$ for agent $i$. Under a Gaussian policy $\pi_\phi$, the next action is computed as follows: $[\boldsymbol{\mu}, \log(\boldsymbol{\sigma^2})] = \pi(s_t) = \alpha(g_{\phi_i}(\bar{o}_{i,t}))$ and $\mathbf{a}_{i,t} \sim \mathcal{N}(\boldsymbol{\mu}, \log(\boldsymbol{\sigma^2}))$, where $g_{\phi_i}$ is a 2-layer MLP taking agent agent $i$'s observation as input and $\alpha$ is the activation function. This induces a parameter-sharing mechanism [19] for policy function making it amenable to generalization.

*Action Interpretation.* External to the policy network, we map the action vector $\mathbf{a}_{i,t}$ for all agents to discrete edge operation set $\bar{e}$ described in Section 2.1, making our approach fully differentiable: We interpret the value of action vector $\mathbf{a}_{i,t} \in \mathbb{R}^d$ as a direct prediction of agent $j$'s embedding with which agent $i$ intends to form a link. It is possible that the action vector of an agent $i$ maps to action vectors of multiple agents $j$. We capture this insight by first stacking action vectors of all the agents into action matrix $\mathbf{a} \in \mathbb{R}^{n \times d}$. We then compute the probabilities of forming a link between two agents as: $\mathbf{a}_{\text{prob}} = \sigma(\mathbf{a}^T \mathbf{a})$, where $\mathbf{a}_{\text{prob}} \in \mathbb{R}^{n \times n}$ is the link probability matrix and $\sigma$ represents sigmoid function. Subsequently, the entire environment structure is updated by modifying the adjacency matrix $\mathbf{A}_t$ as follows: $\mathbf{A}_{t+1}^{ij} = 1$ if $\mathbf{a}_{\text{prob}}^{ij} > 0.5$ and $\mathbf{A}_{t+1}^{ij} = 0$ otherwise. Analogous to game-theoretic approaches – a link is formed/maintained in the next step between two agents only with mutual consent (i.e. both the agents $i$ and $j$ select actions close to each other in the latent space). If either agent do not consent, the link is never formed or severed if it existed in $\mathbf{A}_t$ previously.

*Neural Payoff Mechanism.* We use a lightly structured form of the reward function $r_i$ for each agent $i$, inspiring from the game-theoretic insights where the payoff is designed with an assumption that the agents optimize their position in the network. We use a local 1-layer GNN to compute observation input $(o_{i,(t+1)})$ to the reward function for agent $i$. As in policy function, parameter-sharing makes reward function amenable to generalization. As the actions effectively map to (choice of) agents, we are able to use the state only reward function. Further, the observations for all agents would have been updated based on their actions before computing the reward and hence each agent will have access to other agent's actions and its outcomes (strategy) locally.

## 3.2 Learning

We design an efficient training procedure for learning network emergence games by building on the recently proposed multi-agent adversarial reinforcement learning (MA-AIRL [50]) algorithm. For efficiency, we use multi-agent attention actor critic (MAAC [19]) to solve the inner RL loop of the MA-IRL algorithm. We further account for the graph structured environment by modifying the critic to use graph attention networks [46]. Algorithm 1 in Appendix B outlines complete training procedure. An important consideration for the inverse methods is the extraction of expert demonstrations. Unlike conventional RL environments, where the expert demonstrations are readily available, for observed graphs, we only have access to final graph (experts' outcome). Hence, we need to extract useful and valid trajectories. Following previous multi-agent IRL work, we also extract joint trajectories of each construction in the graph where at each step of the trajectory, we sample an edge for each agent either via random permutation order and use it to define the action of agents. Such expert trajectories will only contain growing graph, but the action space of learning agents still need to consider severance to account for wrong edges constructed they need to remove over time.

## 4 Experiments

In this section, we provide insights into the important aspects of learning network emergence games. First we demonstrate the ability of MINE to discover a payoff mechanism that has high correlation to the ground truth game-theoretic utility and then use a toy real-world network to illustrate that MINE is able to recover the strategic links in the observed network using the learned policy. We further assess the interpretability benefits by qualitatively analyzing the learned reward behavior with respect to the observed network structure. Finally, we evaluate the generalization properties of MINE across different settings. We conclude our experiments by demonstrating MINE's capability to facilitate effective prediction of future strategies (links) of agents given a state of the network. We provide more details on experimental setup in Appendix C and dataset statistics in Table 2(c).

### 4.1 Payoff Function

MINE learns the underlying payoff mechanisms from the observed networks and hence it is important to evaluate its ability to learn a meaningful payoff function useful for interpretation and generalization purposes. Below, we outline our analysis of the learned reward with respect to these properties:

**Quality.** To evaluate the quality of the learned reward, we perform two different experiments: For real-world networks, we do not have access to the ground truth utility that was originally optimized by the involved players. Hence, we first perform a synthetic experiment, where an expert is trained to optimize a specific form of a game-theoretic reward function using the inner MAAC algorithm of MINE (no reward learning). Specifically, we use the following form of the reward inspired from game-theoretical model of social network emergence [51] that considers a trade-off between. benefit and costs of link formation and maintenance to drive the link formation strategies of agents:

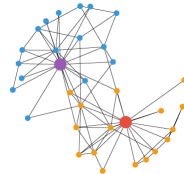

Figure 1: Karate network

$$r_i(o_i) = \sum_{j \in \mathcal{N}(i)} \left( \sum_{k=1}^{K} b_k \max(z_{jk} - z_{ik}, 0) - \|\mathbf{c} \odot (\mathbf{z}_j - \mathbf{z}_i)\|_2 \right),$$

where, $\mathbf{b}$ and $\mathbf{c}$ are benefit and cost parameters respectively with fixed values and $k$ is dimension of agent embeddings $\mathbf{z}$. We perform this experiment with $N = 5$ agents that play the game defined by MINE's MDP but optimize the above reward. We report the correlation (Pearson Correlation Coefficient (PCC)) between the learned and the expert reward and show comparison between ex-

Table 1: Analysis of the learned reward using: **(a)** Game-theoretic reward function **(b)** Zachary Karate club data (no ground truth reward). Correct links are fraction(%) of original links recovered.

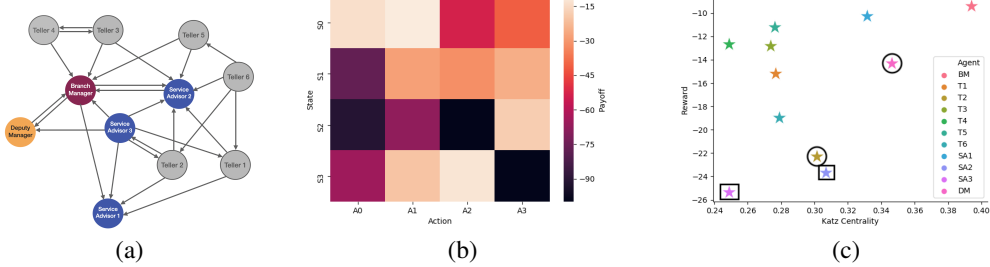

|  | Agent#1 | Agent#2 | Agent#3 | Agent#4 | Agent#5 | Average |
|---|---|---|---|---|---|---|
| PCC | 0.842 | 0.928 | 0.883 | 0.763 | 0.681 | 0.8194 |
| Expert | -7.213 | -12.221 | -10.65 | -6.441 | -12.294 | - |
| MINE | -8.331 | -12.252 | -10.31 | -8.045 | -10.797 | - |

(a)

|  | Leader#Red | Leader#Purple | Community#Red | Community#Purple |
|---|---|---|---|---|
| Correct Links | 72 | 77 | 75 | 81 |
| Original | -132.33 | -83.42 | -99.98 | -74.61 |
| Policy | -141.71 | -85.22 | -112.32 | -77.04 |
| Perturbed | -221.4 | -118.93 | -199.74 | -101.65 |

(b)

(a)      (b)      (c)

Figure 2: Payoff interpretability in relation to the real-world Australian Bank network. **(a)** Observed Network (Node color signify different importance). **(b)** Marginal Payoff heatmap (lighter color signify higher utility) for state-action pairs where state is a single node of particular type and action is the link formation with a new node: (S0): Teller, (S1): Service Advisor, (S2) Deputy Manager and (S3) Branch Manager **(c)** Payoff behavior for an agent w.r.t. its Katz centrality in the network. Circles(Squares) represent increased(decreased) KC of those agents in modified network.

pected returns of the learned and the expert policies. Results in Table 1**(a)** demonstrate that MINE successfully learns a reward function that has high correlation with a ground-truth game-theoretic utility. Further, the learned policies that optimizes this reward imitates the experts well.

Next, we consider a toy real-world network of Zachary's karate club (34 agents, 78 links) that contains two clearly different communities (Figure 1). For learning, we extract expert trajectories as described in previous section. We evaluate the learned policy under following criteria: fraction(%) of correct links recovered (Table 1**(b)** top row) and policy performance in terms of expected return (Table 1**(b)** bottom 3 rows). Community#Red and Community#Purple results are averaged across their follower vertices respectively. The first row demonstrates that MINE recovers a significant portion of strategic interactions (links). To substantiate that this is not the result of mimicking only the structural properties, we report the utility values for 3 different network states: original graph, policy generated graph and perturbed graph, where the perturbation swaps two leaders with their followers. This preserves structural configuration of the graph but results in strategically different network. Table 1**(b)** bottom rows show that network emerging as a result of learned policy has closer behavior to the original graph compared to the perturbed graph under the learned payoff, thereby confirming the vital role of learned objective in recovering real-world strategies.

**Interpretability.** We consider Australian bank dataset [37] (Figure 2(a)), a small network of strategic confiding relationships between branch personnel representing hierarchy among the employees. We first study how the learned payoff can be interpreted with respect to the strategic behavior of agents. After training, we compute marginal utility of individual agents for proposing an action (link formation choice w.r.t other agents). In the heatmap (Figure 2(b)), A0 to A3 signify actions (chosen agents). In real-world settings, agents tend to confine to other agents at the same level or just one level above them in the hierarchy, but rarely with agents at farther level. The heatmap demonstrates similar properties of the learned reward (e.g. a Teller (S0) receives high payoff for proposing a link with other Tellers (A0) and Service advisors (SA) (A1) but not with managers). However, this behavior changes towards the top of hierarchy where clustering behavior is not observed due to fewer agents and confiding relationships become reciprocating with agents at lower level, which is also captured by the learned reward (e.g. Branch manager (BM) (S3) gets high payoff for proposing link with both Deputy manager (DM) (A2) and Sevice Advisor SA (A1)).

We also investigate how the learned payoff relates to the strategic importance of the agents in the network in terms of Katz centrality (KC) [22], a measure also used as part of game-theoretic payoff functions [37]. First, each agent in the observed network (Figure 2(a)) has a separate importance

Table 2: **(a)** Transfer Performance (Top row: transfer across #agents, Bottom 2 rows: transfer across set of agents). **(b)** Strategic Link Prediction Performance: Number are AUC. **(c)** Dataset Statistics.

| Dataset | Correct Links | PCC | Training Episodes |
|---|---|---|---|
| **Andorra** | | | |
| Target Trained Reward (No Transfer) | 76.5 | - | 100000 |
| Source Trained Reward (Transfer) | 70.2 | 0.681 | 72000 |
| Policy Transfer | 68.88 | 0.59 | - |
| **Trade** | | | |
| Target Trained Reward (No Transfer) | 87.2 | - | 25000 |
| Source Trained Reward (Transfer) | 81 | 0.848 | 18000 |
| Policy Transfer | 62.45 | 0.648 | - |
| **Movie** | | | |
| Target Trained Reward (No Transfer) | 62.54 | 0.712 | 60000 |
| Source Trained Reward (Transfer) | 53.09 | 0.631 | 45000 |
| Policy Transfer | 51.1 | 0.598 | - |

(a)

| Datasets | Nodes | edges |
|---|---|---|
| Andorra | 32,829 | 513,931 |
| Trade | 100 | 703 |
| Company | 1984 | 12,751 |
| Movie | 2788 | 10,399 |
| Arxiv GR-QC | 5242 | 14496 |

(c)

| Methods | Trade | Company | Arxiv GR-QC |
|---|---|---|---|
| GT_core | 0.834 | 0.762 | 0.943 |
| Social Game Embed | 0.968 | 0.987 | 0.857 |
| svII | 0.821 | 0.774 | 0.658 |
| Seal | 0.971 | 0.933 | 0.912 |
| Graphite | 0.942 | 0.889 | 0.823 |
| MINE | 0.91 | 0.819 | 0.855 |

(b)

attribute not based on KC. After learning the payoff using the original network, we modify the network such that it affects the KC of particular agents while keeping others same and then compute the state-only utilities for each agent (Figure 2(c)). Agents without circles or squares around them had their KC maintained in the modified network. We observe that tellers with (not-modified) low KC (e.g. T1, T3) get high utility values, explained by peripheral roles of tellers. When we modified the local structure of one teller (T2, golden star in Figure 2(c)) to increase its KC (signified by circles), it gets low utility value owing its overall (low) importance but now high Katz centrality which is contradictory. Further, in spite of having lower KC value for Branch Manager than SA in observed network, BM gets high reward than SA. This shows that MINE captures intricate properties beyond structure that may not be modeled by hand-designed specifications. For instance, KC considers the entire network however, in real-world, an agent often only have access to its local observations. Finally, we modified the observed network to decrease the KC for SA2 (signified by little squares) such that its incoming links are removed. As expected, in spite of having high importance values, SA2 gets a low reward value for not being involved in confiding relationships with tellers.

**Transfer.** In this section, we evaluate the ability of learned payoff to facilitate effective transfer across more player or different set of players with a same game. For all the experiments, we first train our policy and reward functions on source set of agents (that provides expert demonstrations). We then transfer the learned reward function to the target set of agents, where the policy is re-trained to optimize the transferred reward. For comparison, we train the full model on the target set of agents (no reward transfer). Additionally, we perform an experiment to evaluate the generalization capacity of the learned policy. Specifically, we train a full model on source set and then directly evaluate the learned policy on target set without re-training. We report our results on three criteria: fraction of correct links recovered on target network, correlation between the policy performance between target trained model and transferred models (both retrained one and policy transfer) and number of training episodes for convergence. Please note that both source set and target set of agents are extracted from original single network, hence they belong to same game.

*Transfer across more number of agents–* To this end, we consider Andorra phone call network with attributes such as phone type (apple, Samsung and others), location and internet usage. We train MINE on a sub-network of 100 agents to learn the payoff (source training). We then fix the learned payoff and re-optimize the policy over the full network (transfer). Table 2(a) top row demonstrates successful transfer with an on par performance compared to the model fully trained on the large network and notably requires lesser episodes, thereby providing speedup. *Transfer across different set of players–* We consider two strategic networks from different domains: a trade network between countries and bipartite movie network between directors and cast. We spilt each network into two connected components with disjoint set of agents. We train MINE on one component (source training) and transfer it to the other. Table 2(a) showcases highly competitive transfer performance for Trade data, signifying its applicability to extract a useful strategic mechanism from an observed network to train network games with different configurations. For the movie network, the performance degrades slightly and we suspect it is due to its bipartite nature that requires further constraints in the model. Finally, the policy transfer performs worse than re-optimizing the policy on target set of agents. Nevertheless, its moderate success (with no re-training) has applications in scenarios where quick testing may be useful first step. This usefulness of MINE can be attributed to the use of GNN for encoding agent representations that generalize across unseen agents identified by their attributes.

### 4.2 Strategic Prediction

In this section, we evaluate the ability of learned network emergence games as a model to support meaningful strategic predictions. Concretely, we focus on classical link prediction task that forms the basis of many further network analysis tasks. We consider three networks from different domains: a financial trade network, a company network of communication between members at different hierarchy and a co-authorship network of General relativity and Quantum Cosmology field. We split the networks into train (80%) and test(20%) edge sets and train a reward and policy over the training edges. At convergence, we roll-out the evaluation policy asking agents to form links between them. We report the standard link prediction metrics Area under the curve (AUC) in Table 2(b). For baselines, we use 3 game theoretic approaches for link prediction as a direct comparison: GT_core [35] and Social Game Embed [51] that combines network embedding approaches with game-theoretic payoff functions and svII [44], a recently proposed similarity measure used to perform link prediction based on agent similarities. For completeness, we compare with a state-of-art discriminative model SEAL [52] and generative model Graphite [15] for link prediction. The results in Table 2(b) demonstrate that a learned MINE model has reasonable predictive capabilities that outperforms or achieves comparable performance to game-theoretic approaches. The high performance of Social Game Embed for Trade and Company dataset is attributed to its dataset specific payoff function but its performance degrades on dataset which uses a different strategy than a social game while MINE demonstrates consistently good performance. The superior performance of learning baselines for link prediction is expected as these baselines use a task-dedicated architecture and training objective, which stands in contrast to MINE, which discovers the objective from the observed network and learns a generic network emergence strategy model. This demonstrates compelling generalization properties of MINE that is coupled with the interpretability benefits, Finally, none of the above approaches facilitate generalization across new players which makes our approach versatile.

## 5   Conclusion

In this paper, we investigate the problem of learning network emergence games directly from the observed networks without any assumptions on the underlying strategic mechanisms. We propose, MINE, a data-driven learning framework that incorporates Markov-Perfect Network emergence game dynamics into a sequential decision process formulation and solves it using multi-agent reinforcement learning. MINE jointly discovers agents' strategy profiles in the form of learned policy and the latent payoff mechanism in the form of learned reward function. Our experimental evaluation of the predictive, generalization and analytical properties of MINE demonstrates that MINE successfully combines the interpretability benefits of game-theoretic frameworks with the practical applicability of learning approaches. This opens up new avenues for research on leveraging game theoretic approaches to build interpretable and generalizable learning frameworks for network analysis and improving them for scalability. Our current notion of interpretability revolves around analyzing the payoff behavior with respect to observed network properties. An interesting alternative would be to build game-theory inspired learning approaches for network analysis that results in interpretable models themseleves.

## Broader Impact

Networks are ubiquitous structures found across wide range of domains including social, physical, economic systems and many more and any learning approaches in this space has impact on wide ranging applications that consume, process or generate network data. This work focuses on a specific class of networks that come into existence as a result of strategic behaviors adopted by living agents such as humans. Examples of such networks include financial networks, trades between countries, social collaboration networks, human mobility networks and many more, where the network structure itself may be the outcome of direct or indirect actions of human participants. Further, such participants exhibit both bounded rationality and selfishness in their behavior. Analyzing such networks provide insights into the underlying mechanisms that lead to the emergence of such networks. With the increasing availability of network data, learning approaches that can analyze strategic mechanisms would empower AI systems that would help practitioners such as economists, lawmakers and disaster management personnel in the decision-making process. For instance, financial organizations would

benefit by understanding the strategic behavior of market stakeholders and similarly understanding emergence of contact networks would help design effective quarantine policies.

We expect our work to serve as an initial step towards building interpretable approaches to learning such mechanisms from previously available strategic networks and use it for predictive purposes. Also, this line of work can be used to build network design simulators, where the learned mechanism can be used to simulate various network configuration by varying information about the agents which can have ethical considerations. Hence, we caution that this initial attempt will require thorough analysis and follow-up efforts to be put it into practice in real-world settings.

## Acknowledgments and Disclosure of Funding

This work was supported in part by National Science Foundation IIS-1717916 and Shenzhen Institute of Artificial Intelligence and Robotics for Society (AIRS). HZ was on leave from College of Computing, Georgia Institute of Technology.

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
