[Supplementary Material]

# Appendix for Learning Strategic Network Emergence Games

## A   Network Emergence Games and Multi-Agent Inverse Reinforcement Learning

In this section, we first provide details on Markov Perfect equilibrium (MPE) as a solution concept for Markov Network Emergence games and outline how the procedure for solving it maps to the reinforcement learning setting. Next, we discuss the connection of MPE with more general solution concept of Markov Quantal Response Equilibrium (MQRE) that aids in building a gradient based learning approach. Finally, we discuss how the logistic version of MQRE is an appropriate solution concept for solving network emergence games with multi-agent reinforcement learning. We note that several materials discussed in this section can be found scattered across different literature resources cited in the main paper, but we discuss some of these details here in a consolidated manner using canonical notations for ease of exposition and establishing direct correspondence to our approach.

The description of network emergence policy (in Section 2.1) $\pi_i : \mathcal{S} \to \mathcal{P}(\mathcal{A}_i)$, where $\mathcal{P}(\mathcal{A}_i)$ is the distribution over agent $i$'s actions space, concretely specifies the Markov strategy for an agent $i$.

**Definition 1** (Markov Perfect Equilibrium). *A Markov perfect equilibrium $(\pi_i^*)_{i \in N}$ is a Nash equilibrium in Markov strategies.*

While existence of a Markov Perfect Equilibrium (MPE) in stochastic games has been long established for stochastic games [12, 45, 43], an application of Bellman's optimality principle [2] shows that solving for MPE directly maps to recursive procedure of learning optimal joint policy $\bar{\pi}^*$ by optimizing individual reward $r_i$ as done in a reinforcement learning procedure.

**Theorem 1** (Recursive representation of MPE). *A joint Markov strategy profile $\bar{\pi}^*$ constitutes a Markov perfect equilibrium if and only if:*

1. *For all agents $i$, there exist state value function $V_i^* : \mathcal{S} \to \mathbb{R}$ such that:*

$$V_i^*(s) = \max_{a_i \in \mathcal{A}_i} r_i(s, a_i, \bar{\pi}_{-i}^*) + \gamma \sum_{s' \in \mathcal{S}} P_{ss'}(a_i, \bar{\pi}_{-i}^*) \cdot V_i^*(s') \tag{4}$$

   *holds for all states $s_t \in \mathcal{S}$, where $\bar{\pi}_{-i}^*$ represent optimal strategy of all agents other than agent $i$.*

2. *For all states $s_t \in \mathcal{S}$, the joint strategy profile $\bar{\pi}^*$ constitutes a Nash equilibrium of normal-form game (Nash for current state) with action space $\mathcal{A}$ and agent-specific payoff value:*

$$\hat{r}_i(s, \bar{a}) = r_i(s, \bar{a}) + \gamma \sum_{s' \in \mathcal{S}} P_{ss'}(\bar{a}) \cdot V_i^*(s') \tag{5}$$

   *for $\bar{a} \in \mathcal{A}$ and all agents $i$.*

*Proof.* See [8] Page 374 for a proof sketch. $\qquad\square$

In stochastic games such as network emergence, the subsequent course of play which depends on strategies of all players, affects the final outcome in addition to the payoff i.e. decisions are based on:

$$\hat{r}_i(s, \bar{\pi}) = r_i(s, \bar{\pi}) + \gamma \sum_{s' \in \mathcal{S}} P_{ss'}(\bar{\pi}) \cdot V_i(s') \tag{6}$$

As the payoff and strategies are interdependent, MPE can be found by simultaneously solving for state values $\bar{V}^*$ and strategies $\bar{\pi}^*$ using the following maximization operations for all $s_t \in \mathcal{S}$ and all agents $i$:

$$\begin{aligned} V_i^* &= \max_{\pi_i} \hat{r}_i(s, \pi_i, \bar{\pi}) \\ \pi^* &= \arg\max_{\pi_i} \hat{r}_i(s, \pi_i, \bar{\pi}) \end{aligned} \tag{7}$$

Multi-agent Reinforcement learning algorithms are capable of solving the above system of equations in (7) efficiently, where optimizing individual reward functions for learning joint optimal policy ($\bar{\pi}$) maps to solving for Markov Perfect Equilibrium. However, solving directly for MPE requires solving for Nash equilibrium at each state, which is not amenable to learning due to discontinuous characteristics of Nash Equilibrium with respect to payoff matrix. Further, Nash equilibrium assumes all agents to be perfectly rational which is often not the case for agents participating in the network emergence games. Fortunately, both these difficulties are addressed by another solution concept, Quantal Response Equilibrium (QRE) and its logistic version [31, 32], which is stochastic generalization of Nash equilibrium. Specifically, QRE accounts for bounded rationality using a parameter $\lambda$ and models the situations where payoff matrices are injected with some noise, thereby introducing smoothness useful for gradient based learning approaches [25]. In the context of stochastic games, QRE has been extended to Markov version by [5, 10] and referred as Markov Quantal Response Equilibrium (MQRE).

Let the expected payoff from playing action $a$ for agent $i$ in state $s$, given strategies of other players, is denoted by:

$$\hat{r}_i(s, a, \bar{\pi}) = r_i(s, a, \bar{\pi}) + \gamma \sum_{s' \in \mathcal{S}} P_{ss'}(a_i, \bar{\pi}_{-i}) \cdot V_i(s') \tag{8}$$

In the quantal response framework, agent $i$ is assumed to perceive noise injected payoff version of Eq. (8) as:

$$\tilde{r}_i(s, a, \bar{\pi}) = \hat{r}_i(s, a, \bar{\pi}) + \varepsilon_i(s, a) \tag{9}$$

The noise vector for all actions $\varepsilon_i(s) = (\varepsilon_i(s, a))_{a \in \mathcal{A}_i}$ follows a joint distribution with zero mean and density function $f_i(\varepsilon_i(s))$. Let agent $i$'s response set of action $a$ in state $s$ denoted by:

$$R_i(s, a) = \left\{ \varepsilon_i(s) \in \mathbb{R}^{|\mathcal{A}_i|} : \tilde{r}_i(s, a, \bar{\pi}) > \tilde{r}_i(s, a', \bar{\pi}) \ \forall a' \in \mathcal{A}_i \right\} \tag{10}$$

$R_i(s, a)$ specifies the realization of $\varepsilon_i(s)$ such that agent $i$ in state $s$ perceives action $a$ as the one with the highest payoff.

**Definition 2** (Markov quantal response equilibrium). *A Markov quantal response equilibrium is a strategy profile $\bar{\pi}^*$ such that*

$$\bar{\pi}^* = \int_{R_i(s,a)} f_i(\varepsilon_i(s)) \, d\varepsilon \tag{11}$$

where the probability mass function of the response set of agent $i$ specifies the probability that agent $i$ in state $s$ takes action $a$. An interesting version of MQRE is its logistic version (MLQRE) which arise from the noise that is i.i.d according to Gumbel distribution with parameter $\lambda \in \mathbb{R}_0^+$, which also controls the rationality of agents. An MLQRE $\bar{\pi}^*$ can then be expressed in closed form as a solution to the following system of equations: For all states $s_t \in \mathcal{S}$, all agents $i$ and all actions $a \in \mathcal{A}_i$:

$$
\begin{aligned}
\bar{\pi}_i^*(a_i|s) &= \frac{e^{\lambda \cdot \hat{r}_i(s, a, \bar{\pi}_{-i}^*)}}{\sum_{a' \in \mathcal{A}_i} e^{\lambda \cdot \hat{r}_i(s, a', \bar{\pi}_{-i}^*)}} \\
V_i^*(s) &= \sum_{a' \in \mathcal{A}_i} \bar{\pi}_i^*(a_i|s) \cdot \hat{r}_i(s, a', \bar{\pi}_{-i}^*)
\end{aligned} \tag{12}
$$

When $\lambda \to 0$, the equilibrium is fully noisy and the agents will select actions uniformly at random. When $\lambda \to \infty$, the agents will choose actions in best response manner (greedily). Recently, [10] has shown that if the logit MQRE converges as $\lambda \to \infty$, the limit point is Markov perfect equilibrium.

**Theorem 2.** *Let $\bar{\pi}^{lim}$ be the limit point of some logit MQRE $\bar{\pi}^*(\lambda)$, i.e.*

$$\bar{\pi}^{lim} = \lim_{\lambda \to \infty} \bar{\pi}^*(\lambda) \tag{13}$$

*Then, $\bar{\pi}^{lim}$ is a Markov perfect equilibrium.*

*Proof.* See [10] Page 13, Theorem 4 for a proof by contradiction. $\qquad\square$

[10] further provides proof on the existence of the limit point for stochastic games which establishes logit MQRE an appropriate equilibrium concept for characterizing trajectory distributions induced by the reward functions specified in the decision process of Markov Perfect network emergence games. However, the system of equations in Eq. (12) provide a solution to a set of constraints which do not explicitly specify concrete joint policy profiles that can be used to maximize the likelihood of the observed data as a function of rewards, a key objective of this work. To address this exact challenge, [50] introduces a new solution concept, referred as Logistic Stochastic Best Response Equilibrium (LSBRE), that corresponds to the outcome of repeated application of a stochastic process where each agent attempts to optimize its actions while keeping other agents' actions fixed. Please refer to Section 2.3 for the complete definition of LSBRE for a Markov game with horizon $T$ - a key component of which is the form of $z_i$ which is given as:

$$z_i^{t,(k+1)}(s^t) \sim P_i^t(a_i^t | \bar{a}_{-i} = \bar{z}_{-i}^{t,(k)}(s^t), s^t) = \frac{\exp\left(\lambda Q_i^{\bar{\pi}^{t+1:T}}(s^t, a_i^t, \bar{z}_{-i}^{t,(k)}(s^t))\right)}{\sum_{a_i'} \exp\left(\lambda Q_i^{\bar{\pi}^{t+1:T}}(s^t, a'^t_i, \bar{z}_{-i}^{t,(k)}(s^t))\right)} \quad (14)$$

where, $Q_i^{\bar{\pi}^{t+1:T}}$ is recursively defined as:

$$\begin{aligned} Q_i^{\bar{\pi}^{t+1:T}}(s^t, a_i^t, \bar{\pi}_{-i}^t) = \hat{r}_i(s^t, a_i^t, \bar{\pi}_{-i}^t) + \mathbb{E}_{s^{t+1} \sim P(\cdot|s^t, \bar{a})} \big[ \mathcal{H}(\pi_i^{t+1}(\cdot|s^{t+1})) \\ + \mathbb{E}_{a^{t+1} \sim \bar{\pi}^{t+1}(\cdot|s^{t+1})} [Q_i^{\bar{\pi}^{t+2:T}}(s^{t+1}, \bar{a}^{t+1})] \big] \end{aligned} \quad (15)$$

The reward term $\hat{r}_i(s^t, a_i^t, \bar{\pi}_{-i}^t)$ in Eq. 15 corresponds to the payoff term $\hat{r}_i(s, a, \bar{\pi}_{-i}^*)$ in Eq. 12. Further, [50] establishes that the trajectory induced by LSBRE policies can be characterized with energy-based formulation such that the probability of a trajectory increases exponentially as the sum of rewards increases. This allows them to build a practical inverse reinforcement learning algorithm for multi-agent setting where the expert policies are assumed to form a unique LSBRE under some unknown reward. We build our approach based on this algorithm and extend it to network emergence games which we discuss in the next section.

# B MINE Algorithm

In this section, we outline the multi-agent inverse reinforcement learning procedure to learn network emergence games.

---

**Algorithm 1** MINE Algorithm

---

1: **procedure** MINE
2:    **Input:** Expert Demonstrations $D_E = \{\tau_j^E\}$, Empty list of Agent's trajectories $D_\pi$,
3:    Replay Buffer $\mathcal{B}$, Agent feature matrix $\mathcal{X}$
4:    Initialize the parameters for policies $\bar{\pi}$, attentive critic $\bar{Q}$, reward estimators $\bar{g}$, potential
5:    functions $\bar{h}$ and state encoder (SE) with $\bar{\phi}, \bar{\psi}, \bar{\omega}, \bar{\theta}$ and $\varphi$ respectively
6:    **repeat**
7:       Reset the environment and set adjacency matrix $\mathbf{A}_0 = \mathbf{0}$ (no links)
8:       Initialize $\tau^\pi = \{\}$
9:       **for** each step do **do**
10:          Sample $\bar{a} \sim \bar{\pi}_{\bar{\phi}}(\bar{a}_t | \bar{o}_t)$
11:          Update $\mathbf{A}_t$ based on $\bar{a}$ (c.f. Section 3)
12:          Update $s_{t+1} = \mathsf{SE}_\varphi(s_t, \mathbf{A}_{t+1})$
13:          Add $(\bar{o}, \bar{a}, \bar{o'})$ to $\mathcal{B}$
14:          Update $\tau^\pi \leftarrow \mathsf{concat}(\tau^\pi, (\bar{o}, \bar{a}, \bar{o'}))$
15:       **end for**
16:       Add $\tau^\pi$ to $D_\pi$ and reset $\tau^\pi \leftarrow \{\}$
17:       **for** each training iteration do **do**
18:          Sample $(\bar{o}, \bar{a}, \bar{o'})_\pi$ triples $T_\pi$ from $D_\pi$ and $(\bar{o}, \bar{a}, \bar{o'})_E$ triples $T_E$ from $D_E$
19:          Update $\bar{\omega}, \bar{\theta}$ using:
20:          $\mathbb{E}_{T_E}[\log D((\bar{o}, \bar{a}, \bar{o'})_E)] + \mathbb{E}_{T_\pi}[\log(1 - D((\bar{o}, \bar{a}, \bar{o'})_\pi))]$
21:          Update reward estimates $\bar{r}(\bar{o}, \bar{a}, \bar{o'})$ with $\bar{g}_{\bar{\omega}}(\bar{o}, \bar{a})$
22:
23:          Update $\bar{\phi}, \varphi$ w.r.t. $\bar{r}(\bar{o}, \bar{a}, \bar{o'})$ using MAAC policy gradients:
24:          $\nabla_{\phi_i} J(\bar{\pi}_\phi) = \mathbb{E}_{\bar{o} \sim \mathcal{B}, \bar{a} \sim \bar{\pi}}[\nabla_{\phi_i} \log(\pi_{\phi_i}(a_i | o_i))(-\alpha \log(\pi_{\phi_i}(a_i | o_i))$
25:                $+ Q_{\psi_i}(\bar{o}, \bar{a}) - b(\bar{o}, a_{-i}))]$
26:
27:          Update critic parameters $\bar{\psi}$ by minimizing the TD-error:
28:          $\mathbb{E}_{(\bar{o}, \bar{a} \sim B}[\sum_{i=1}^{N}(Q_{\psi_i}(\bar{o}, \bar{a}) - y_i)^2]$ where
29:          $y_i = r_i(\bar{o}, \bar{a}) + \gamma \mathbb{E}_{a' \sim \bar{\pi}_{\bar{\phi}}(\bar{o'})}[Q_{\psi_i}(\bar{o'}, \bar{a'}) - \alpha \log(\pi_{\phi_i}(a_i' | o_i'))]$
30:       **end for**
31:    **until** Convergence
32:    **Output:** Learned policies $\bar{\pi}_\phi$, reward functions $\bar{g}_\omega$ and state encoder $\mathsf{SE}_\varphi$
33: **end procedure**

---

Algorithm 1 builds on the recently proposed multi-agent adversarial reinforcement learning [50] technique by extending it to support graph structured environment and modifying it to use multi-agent attention actor-critic (MAAC [19]) algorithm in the inner RL loop for efficient off-policy learning. At the start of each epoch, the game is reset such that there exists no links between the agents (line 7, $\mathbf{A}_t$ is the adjacency matrix at step $t$). At this stage, agents' low-dimensional embeddings are initialized from their features using one round of the the GNN based state encoder (due to lack of edges there will be no message passing initially). Next, we rollout several forward steps to collect experience in replay buffer (line 13) and build agents' trajectories (line 14). During each step in the game, we sample actions for all agents using our Gaussian policy (line 10, corresponds to each agents' announcements of forming links with other agents as discussed in Section 3). These continuous actions are mapped (externally to the network) to links between agents and used to update $\mathbf{A}_t$. Based on the new structure, the state of the environment is updated using SE (line 12). After several rollouts, few iterations of gradient updates are used to update the parameters of all the networks. For each update, we obtain set of trajectory samples from both expert and agents' trajectories (line 18). For the loss function in line 20, we follow [50] and use the following structure for the discriminators: $D_i(o_i, a_i, o_i') = \frac{\exp(f_{\omega_i}(o_i, a_i, o_i'))}{\exp(f_{\omega_i}(o_i, a_i, o_i')) + \pi_{\phi_i}(a_i | o_i)}$, where $f_{\omega_i}(o_i, a_i, o_i') = g_{\omega_i}(o_i, a_i) + \gamma h_{\theta_i}(o_i') - h_{\theta_i}(o_i)$. $g_\omega$ is the reward estimator and $h_\theta$ is the potential function. We use this discriminator definition to

update parameters $\bar{\omega}$ and $\bar{\theta}$ (line 19-20). The updated reward estimator $g_\omega$ serves as an updated reward function $\bar{r}$ (line 21). The parameters of the structured strategy network (which includes both the policy network and the state encoder network) are updated with respect the newly estimated reward function $\bar{r}$ using the soft-policy gradients (line 24). Our updates are based on off-policy gradients as we use MAAC [19] for policy learning that extends Soft-Actor Critic [16] algorithm to multi-agent setting with attention based critics. In line 24-25, $\alpha$ is the temperature parameter that balances the trade off between policy entropy and reward maximization. Further, $b$ is the baseline advantage function used to solve the credit assignment problem in multi-agent setting and we follow [19] to compute it. Finally, the attentive critics are updated by minimizing the TD-error as shown in line (28-29). We note that there is no explicit objective function for directly optimizing the state encoder parameters $\varphi$, however, the policy gradients backpropagate through the state encoder network in an end-to-end fashion, thereby updating the encoder network. At convergence, MINE returns the learned policy network $\bar{\pi}$, reward function $\bar{g}$ and state encoder SE, which are then used for performing evaluation tasks as described in next section.

## C Further Experiment Details

### C.1 Datasets

In this section, we provide more details on the properties of the datasets used in our work.

**Andorra**[1]. The Andorra dataset contains call detail records between 32,829 citizens where a link between two citizens if they were involved in atleast one call interaction during the period from July 2015 to June 2016. The dataset contains three attributes for each agent – Phone type (takes values Apple, Samsung, others), frequent city (takes values between 0 and 6) and cellular usage (real value). These attributes are strongly correlated with important unobserved individual characteristics such as phone type may be related to income, frequent city to the place of dwelling and cellular usage to the daily online activities.

**Trade**[1]. This is the 2014 international trade data between countries provided by the United Nations Statistical Division (UN Comtrade Database: https://comtrade.un.org/). The network contains 100 countries, a link among which indicates that the trade value between two countries is greater than 1 billion dollars in both directions. the dataset contains three attributes – Continent (Africa, America, Asia/Pacific and Europe), Economic Complexity Index and GDP (real value). While continent affects the location based strategic trade decision, ECI captures the diversity and sophistication of country's export and GDP directly impacts a country's ability to perform trade partnerships. In this dataset, an entire country represents an agent in the game and hence this is an example where the network emerges due to indirect impact of human beings.

**Movie**[1]. The movie dataset is specific type of social network where edges signify collaborations between directors and actor/actresses (cast). The links in this network are formed strategically based on the benefits to both parties. The overall network structure is close to bipartite network (some nodes are both directors and cast members) with 160 directors, 2628 cast members and 10,399 links between them. The dataset was collected for 3493 movies in the period of 2000-2016 and contains two attributes for each agent – Occupation (director, cast) and Gender (male, female).

**Company**[1]. This data consists of network between employees in a company where an edge between them signify a call or text communication. Each employee is either a manger or a subordinate (which is also the only attribute for agents in this network) and there are 420 managers and 1564 subordinates with 12,751 edges between them. In this network, managers are mostly connected to other managers and similar for subordinates with occasional links between subordinates with their specific manager. This is an example of a hierarchical network that the properties with the Australian bank toy network that we used to analyze the reward interpretability.

**Arxiv GR-QC**[2]. This is a collaboration network between authors in the field of General Relativity and Quantum Cosmology where an edge between them indicate they co-authored atleast one paper between them. The network consists of 5242 nodes with 13396 edges between. This network does not contain any attributes hence we only use one-hot identity map as initial feature vector for agents. In this network, the strategy of link formation between two authors would depend on their common neighbors and hence the network emerges based on development of its own structure.

## C.2 Baselines

In this section, we provide details on baselines used for comparison in strategic prediction task.

**GT_core** [35]. This method incorporates game-theoretic models into node representation learning methods for learning latent node features that are used for downstream link prediction. Specifically, this method focuses on two node representation learning methods – node2vec and DeepWalk and enhances them by proposing a novel form of biased random walk. In this biased walk, the next node is not chosen randomly, instead it is chosen based on link probability between the current node and set of possible next nodes. The link probability is computed as a product of two weight values corresponding to that edge. These two weight values are computed using game-theoretic model based utility function and k-core decomposition respectively. The game-theoretic weights are further computed using two different utility models - a co-authorship model [21] and influence games [33]. Once the biased random walk is performed to select the neighbors, standard gradient based training is done and link prediction is performed based on learned node representations.

**Social Game Embed** [51]. This work takes an economic view of link formation between heterogeneous agents, where the link formation is considered to be driven by the tradeoff between exchange benefits and coordination costs between interacting agents. Based on this view, a social network formation model is proposed with utility function designed to capture this relation between benefits and cost. The agents in the network are characterized by vectors, called endowment vectors and agents are assumed to maximize their utility by comparing their endowment vectors with those of others. This vectors are learned from the observed network by solving an optimization problem. Following is the form of the utility function of agent $i$ with respect to agent $j$: $\sum_{k=1}^{K} b_k \max(z_{jk} - z_{ik}, 0) - \|\mathbf{c} \odot (\mathbf{z}_j - \mathbf{z}_i)\|_2$, where the parameters $\mathbf{b}, \mathbf{c}, \mathbf{W}$ are learned by minimizing the loss function $\mathcal{L}(\mathbf{b}, \mathbf{c}, \mathbf{W}|D)$. Note that the hand-designed reward function used for the synthetic experiments in our work is inspired from the above function. Once the parameters are learned, link prediction is performed by using the utility of an agent $i$ with respect to other agents $j$ as a predictor.

**svII** [44]. svII is a game-theory based interaction index that captures the notion of similarity between agents. The interaction index is build upon two well-known solution concepts from game theory - the Shapely value and the Bazhaf index. The payoff for an agent in the network is a function of its sphere of influence parameterized by k, where k is the degree of influence. Given this influence game, the above index measures similarity between the spheres of influence of any two nodes. Link prediction is performed by computing the similarity between any pair of nodes not having an edge in current graph and connect the most similar pair.

**SEAL** [52]. This is the state-of-art discriminative model for link prediction that uses graph neural network to learn general graph structure features from local enclosing subgraphs, embeddings and attributes, thereby capturing higher order properties. Link prediction is performed using standard technique used in learning based approaches.

**Graphite** [15]. Graphite is a state-of-art latent variable generative model for graphs based on variational auto-encoding. Graphite learns a directed model expressing a joint distribution over the entries of adjacency matrix of graphs and latent feature vectors for every node. Graph neural networks are used in straightforward manner for inference (encoding), while the decoding of these latent features to reconstruct the original graph (generation) is done using a multi-layer iterative procedure. Link prediction is performed by first training the model on a subgraph, adding the test edges back to the graph and evaluating the probabilities assigned to the test edges.

## C.3 Evaluation Protocol

In this section, we elaborate and clarify the evaluation protocol used in the modified (perturbed) network setting (quality and interpretability tasks in Section 4) and strategic prediction setting. As described in Algorithm 1, at the start of every epoch, the environment is reset to an empty graph state (only agents, no links) and then the agents learn to form links by observing the real network. However, for evaluation purposes, it is not required to start from an empty graph state and at convergence, one can input a graph with edges as initial input to MINE and perform evaluation tasks thereof. Note that both the policy network and reward network employ a GNN based architecture and hence any graph structure provided as input will be encoded into continuous representation by these GNN before

being used as input to either policy or reward network. We outline the evaluation steps for each task below:

**Quality.** For the experiments on Karate network (Table 1(b)), the values for three different graphs are the agent specific utilities for two leaders and average of agent-specific utilities across all agents of each community. To obtain these numbers, we first train MINE using the observed network data. Once the model is trained, we input the three graph configurations to the learned reward model and compute the agent-specific utilities directly. The perturbation is done on the original network and the policy generated graph is obtained by running the evaluation policy starting from an empty network.

**Interpretability.** For the experiments on Australian bank dataset with respect to Katz centrality, we first train MINE using the original network. We then modify the original network to alter centrality of some nodes. Next, we provide this new network as input to the learned reward function without running the evaluation policy. Note that reward estimator is a learned function that takes agent-specific local observation structure (encoded via GNN) as input and hence it directly processes the input network observations for each agent and return the reward value that is used to report the analysis.

**Strategic Prediction.** For this task, after the training ends, we provide the graph with 80% training edges as an initial input to MINE. Next we run the evaluation policy and collect the new links formed by the agents. We use these links as predictions and compare them with the test edges to report the results.

## C.4    Training Configurations

We design our approach based on recently proposed MA-AIRL algorithm but replace its inner loop on-policy RL algorithm with off-policy MAAC algorithm. Hence, we closely follow the architecture and training configurations of MA-AIRL for the descriminators while we adopt the configurations of MAAC for policy learning. MAAC uses Soft-Actor critic to update the policies and in general, SAC does not have any aggressive hyper-parameters that needs to be tuned. We use a target critic function $Q_{\hat{\psi}}$ following [19] as it helps to stabilize the use of experience replay for off-policy reinforcement learning with neural network function approximators. For GNN, we tune propagation round $P$ between 2-4 but found 2 to be best choice. We tune the node embedding dimension in the range of $\{32, 64, 128, 256\}$ based on the graph. We reset each environment after every 100 steps. After 100 steps, we perform 4 updates for the attention critic and for all policies. We perform gradient descent on the Q-function loss objective, as well as the policy objective, using Adam optimizer. Further, following [19], we use 4 heads for our attention critics and dimension of 128 for all hidden units across networks. Table 3 provides list of hyper-parameters that were used across all the experiments:

Table 3: Hyper Parameter Configuration Table

| HyperParameters | Values | HyperParameters | Values |
|---|---|---|---|
| discount factor | 0.99 | replay buffer size | 1e6 |
| batch size | 1024 | policy and critic learning rate | 0.001 |
| policy entropy temperature | 0.01 | target policy and critic update rate | 0.005 |
| discriminator entropy temperature | 0.01 | discriminator learning rate | 0.0005 |

All the experiments were conducted on Intel Xeon CPU V4  2.20 GHZ with 64 GB memory and Nvidia GeForce 1080 GPU.

# D    More Related Work

In this section, we discuss more literature on equilibrium concepts in network emergence games and inverse reinforcement learning.

**Solution Concepts and Network Emergence Games.** Network emergence games focus on analyzing the construction of equilibrium networks where no agent want to locally change the network [38]. Various equilibrium concepts (with special focus on network stability (robustness)) have been proposed and studied to analyze the formation process in networks. It has been shown that pure Nash equilibrium is a weak and restrictive concept, for instance empty networks (where no agent announces

any link) are always in Nash equilibrium, which further requires that agents' actions in each state to be independent [3]. As a more useful solution concept, [20] proposed pairwise stability that searches for networks that are robust to one link deviation, where link creation is bilateral and under mutual consent of the agents while link severance is unilateral. Building upon two concepts, [20, 34] proposed pairwise-Nash equilibrium that allows for unilateral multi-link severance in addition to mutual one-link creation and thereby effectively model non-cooperative games. [34]'s proposal is a noncooperative linking game in which agents independently announce which bilateral links they would like to see formed and then standard game-theoretic equilibrium concepts apply for making predictions. Finally, [4] proposed another linking game where players can §offer or demand transfers along with the links they suggest, which allows players to subsidize the emergence of particular links. Network emergence games have been generally modeled as one-shot normal form game or an extensive form game and only recently have been investigated by [24] in Markov setting. Finally, couple of approaches[40] have focused on estimating network emergence games where they use data to estimate the parameters of the model. While being computationally inefficient and limited in their practical applications, these approaches demonstrate the promise in investigating the use of observed data to learn the specifications for game theoretic approaches.

**Inverse Reinforcement Learning.** As stated before, the MaxEnt IRL framework [55] aims to recover a reward function that rationalizes the expert behaviors with least commitment and denoted as IRL($\pi_E$): IRL($\pi_E$) = $arg \max_{r \in \mathbb{R}^{S \times A}} \mathbb{E}_{\pi_E}[r(s,a)] - RL(r)$ where $RL(r) = \max_{\pi \in \Pi} \mathcal{H}(\pi) + \mathbb{E}_\pi[r(s,a)]$. Here, $\mathcal{H}(\pi) = \mathbb{E}_\pi[-\log \pi(a|s)]$ is the policy entropy. The forward RL problem in the inner loop makes the above procedure less efficient and hence various improvements have been proposed in the literature. [13, 14] propose one such framework, Adversarial IRL, a sampling based approximation to MaxEnt IRL that uses adversarial training framework where the discriminator takes following specific form: $D(s,a) = \frac{\exp(f(s,a))}{exp(f(s,a)+q(a|s)}$, where $f(s,a,s') = g(s,a) + \gamma h(s') - h(s)$. The policy is trained to maximize $[\log D - \log(1-D)]$ and the specific form of $f$ is used to alleviate the reward shaping ambiguity where many reward function can explain an optimal policy. It has been shown that at optimality, either $f$ or $g$ will approximate the advantage function of the expert policy and thereby recover the reward upto some approximation while $q$ will approximate the expert policy. Our approach is built on [50] that extends the above setup to multi-agent case and we have discussed all the relevant details in the earlier sections. The above method is general in a sense that it supports cooperative, competitive and mixed environments hence useful for our case. There are other inverse learning methods specific to various multi-agent settings and tasks that include cooperative inverse reinforcement learning [17], non-cooperative inverse reinforcement learning [53] and competitive multi-agent inverse reinforcement learning with suboptimal demonstrations [49]. [36] is a very early work in multi-agent inverse reinforcement learning that form the basis of recent advancements in multi-agent inverse reinforcement learning.

## Footnotes

[1] Dataset can be downloaded from repository for [51] at https://github.com/yuany94/endowment/tree/master/data

[2] Data is available at: https://snap.stanford.edu/data/ca-GrQc.html