[Reviews · NeurIPS 2020]

Review 1

Summary and Contributions: The paper introduces a multiagent network formation algorithm based on game-theoretic principles and inverse reinforcement learning to model the appearance, retainment and disappearance of network structures and reconstruct the payoffs. The algorithm shows some robustness which appears to improve beyond more generic algorithms applied in the domain.

Strengths: The paper's model is adapted to the specific domain and therefore takes stronger account of the specificities of the network game model. The model is based on a Markovian treatment of the sequence of networks, on evaluating the "networked" preferences and matching the modeled preference dynamics to the actually observed one.

Weaknesses: While I am not a domain expert, I regularly read game-theoretic and network dynamics papers, yet I found the present paper difficult to read. I suspect that the paper probably tries to convey so much material that it is affected by the space limitation at NeurIPS. It's not necessarily badly written, but it is simply too dense and takes to little time to develop ideas and motivation. I believe that the paper would better be placed in a journal with less space limitation and where the authors can expand properly on context and ideas; I do not think that NeurIPS is the best place to communicate this. I also didn't find the choice of network examples in Figure 2 particularly convincing or transparent. These are two very small networks and it was not clear to me what the rationale of the choice of these networks is and what insight these examples transport. I cannot say that the appendix helped clarifying this. That being said, maybe this paper needs really a deep domain expert for review, but as the conference has a more general audience, I believe the paper should be sufficiently accessible also to a generally informed scientific public. # Edit after rebuttal/discussion I haven't changed my opinion.

Correctness: As far as I could judge, the paper looks correct. On line 643, appendix B, a bracket is missing in the numerator.

Clarity: As mentioned under weaknesses, the paper could profit from a significant expansion, motivation and development to be more accessible. It is very densely written and important decisions are not always motivated. Just as one prominent example: the reward function on line 278 is not motivated, just mentioned and referred to in literature. What is the circle with the center point? Component-wise multiplication? With respect to such things the paper needs to be self-contained. Note that this is not even explained in the supplementary material (but it should be explained in the main text, anyway). This looks more like material for a journal paper condensed into a NeurIPS submission.

Relation to Prior Work: Looks fine.

Reproducibility: No

Additional Feedback: - line 2: non-inanimate - sounds overwrought, why not just "animate"? If there is a reason, explain. - line 53: sentence grammar: "have proposed Markov version" - article missing - line 98: where did the A disappear to in P_T(S)? - line 111: the notation does not seem consistent. G = (V,A,X) is an element of G, but in the previous line, G has been defined as a set symmetric zero-diagonal matrices (or else undirected edge sets). Which of these? If they are equivalent, explain. - line 113: what is the feature set going to model? - line 116: A_0 = 0 means no connections? - line 121: what is d referring to? The neighbours that a vertex can see? - line 164: space should be after not before colon - line 164: the very last seems to have been a \pi, but lost the backslash. - line 165: "to perceive noise injected" -> "to perceive the noise-injected" - line 194: what's k? Number of iteration steps? - line 215: there is too much information packed in this line. What is the message doing, what is it transporting, what is U evaluating, etc. - line 222: "Gaussian policy" - what space does this live in. The actions, I thought were creation, maintenance and deletion of links. - line 230: how are the actions of the two agents combined? I found the whole process of how agents' action matrix operates difficult to parse. - Figure 2: (a) you talk about darker nodes, but all I see are different colours. Different colours cannot be easily compared with respect to their brightness. What does (b) tell us? What are the little circles in (c) indicating and how should we read/interpret the result? - Figure 1: why does it come after Figure 2? - I had to spend time looking for it. - line 299: this statement seems to be important. Are you saying that the advantage of your method is due to the learned policy and that is more effective than learned objectives? If so, and I understood this correctly, this needs to be a prominent tagline of your work and your arguments more built up to make this point. - line 329: what do you keep fixed when you transfer the learned reward function? - Table 2 (a): What numbers is the first column showing? Percentages?


Review 2

Summary and Contributions: This work seeks to estimate payoffs and policies that jointly form a Markov perfect equilibrium for sequential network formation games, with the aim of predicting the future evolution of real-world networks (of both the same class, and through transfer learning, of different classes). The main contribution is a learning framework using graph neural networks (allowing for abstraction away from the details of number of agents and exact network structure) for both the policy and reward models. This model is evaluated on both synthetic and real-world data.

Strengths: This work considers an important area. There is a lot to like about its approach: Learning payoff functions directly from the data is preferable to specifying an ad hoc model (as the paper convincingly argues). The use of GNNs to learn the policies and payoff functions is an intriguing direction.

Weaknesses: There are a number of errors in the preliminaries section, both minor and substantive; see "correctness" for details. I found the paper difficult to understand in many places; see "clarity" for details. Overall, I think the project is worthwhile and seems to have merit; however, I cannot recommend that this paper be accepted in its current form. This paper packs a lot into eight pages, but it's not always clear that it has demonstrated its claims. "Interpretability" appears to be a central claimed example, but it's not clear what is meant by that in the context of neural models. One aspect of the empirical evaluation that I am concerned by is the dependence on the final form of the network. In training, actual edges are added to the network is essentially random order (since the real order is not available); this makes it hard to validate the learned model.

Correctness: I am not convinced by some of the decisions made for the empirical evaluation (the need to infer connection order described in "weaknesses" primarily). Other parts of the evaluation simply don't make sense to me; they may be sensible, but I don't understand how the claims follow from the presented evaluation (see the third item in "clarity"). Beyond the notation issues described in "clarity", the preliminaries sections contained several a number of definitions that were either errors or needed further clarification: * p.3 l.108: The expression for the expected discounted utility starting from time t is off by a constant factor (namely, \gamma^{t-1}) * p.4 l.162: V^*_i(s) as defined is simply the constant value 1 at all states s; this is surely not what you meant. Also the sum ranges over a', which is never used within the sum; I assume a_i is meant to be a'. * p.4 l.166: "In the quantal response framework, agent i is assumed to perceive noise injected payoff version of this expression a \hat{u}_i(...) = u_i(...) + \epsilon_(s,a)...": This is true but not clearly related by the text to the form of the policy described above. The reason that agents behave stochastically is precisely because they are deterministically best-responding to a stochastically perturbed utility. * p.4 l.186: subtracting RL(r) seems to minimize entropy and/or max attainable reward for a given reward function, which seems curious for a max-entropy approach. Can you give some intuition about what is going on here?

Clarity: The paper is not as clear as it needs to be. Critical aspects of the architecture and datasets are described in a way that feels rushed and incomplete. I did not feel that enough detail was provided for me to thoroughly evaluate the paper's claims. I list four major clarity issues below, followed by a number of notation issues and other minor problems. * p.5 l.236-242: The description of the reward function is extremely brief and confusing. Are we learning a separate reward function for each agent? If so, how does transfer work? The text seems to suggest that reward is a function of state only, but I don't understand how. * Similarly, the policy function conditions on player-specific parameters \phi_i; how could this work for transfer when the test set consists of different agents? * p.7 l.312-325: I found this paragraph on the relationship between learned payoffs and Katz centrality very confusing; I don't understand the checks being performed here or what they are in aid of. * s.4: I'd like to hear a lot more about what these datasets actually measure. E.g., the bank dataset purports to measure "confiding relationships"; what are those and how would you ever measure them? Notation issues / minor issues - p.3 l.97: The notation for distributions is inconsistent and potentially confusing; e.g., the use of P_T as a function of state and action profile, as a set of distributions over states, and as a function of state alone, is confusing. I suggest using explicit notation such as $\Delta(X)$ (the set of all distributions over discrete set $X$) where appropriate. - Why is u_i in bold? - It would be better to choose different letters for adjacency matrix \mathbf{A} and actions sets A_i rather than distinguishing them by typeface alone. - p.3 l.131 "we do not impose any specific functional form on the reward function", and p.3 l.135: "we outline the parameterization of the reward function in the next section": These two statements are not consistent with each other. I think you mean that you do not make strong structural assumptions about the reward function, and instead use a very flexible GNN functional form for the reward function. - p.5 l.221: What is \bar{o}_{i,t}? Why is a quantity with a bar (representing profile of something) being indexed by user? - Table 2 needs much better captions. (a) is labeled only as "transfer performance", but the dataset and prediction tasks are not described. Even though some of this information is given in the text, a table or figure plus its caption should ideally make sense standing alone.

Relation to Prior Work: A good survey of related work is included as part of the introduction.

Reproducibility: No

Additional Feedback: == post-feedback comments == The author response has satisfactorily addressed my concerns about correctness. My concerns about clarity remain, but with the extra page for camera-ready it should be possible to add some missing details. I've increased my overall score accordingly. One overall suggestion would be to lean a little less on the expertise of the reader. A number of responses to my clarity questions pointed to presentations or datasets being standard in various disjoint communities. It is unlikely that your readers will be experts in each of these communities; your paper will have considerably more impact if you make your justifications explicit. This doesn't need to take a lot of space! If, for example, the definition given of IRL is standard, a single phrase and/or citation indicating this would go a long way.


Review 3

Summary and Contributions: Propose MINE, a framework to learn network formation games as a way to predict or explain network structure. The approach assumes a structured form of network emergence game, and uses a form of multiagent iRL to induce payoff data fitting a QRE model. The goal is to jointly discover agents’ strategy profiles (network emergence policy) and the latent payoff mechanism in the form of learned reward function, and then use this to account for observed structures or make predictions.

Strengths: Elaborate and conceptually well-founded approach for combining data and strategic modeling for network modeling. Thorough experimental analysis covering a range of examples and issues.

Weaknesses: Complex methods combining a plethora of techniques. Difficult to unpack the underlying assumptions and contributions of different elements of the model.

Correctness: Claim 1: MINE payoff mechanism highly correlates with the ground truth. • Use synthetic experiment with 5 agents and Karate club network to demonstrate • Uses the fraction of links recovered as a metric, but would like more information about the links ◦ Did MINE just miss recovering some links or could it also have links that don’t exist in the actual network and what are the statistics on the latter? ◦ Depending on application, could be important Claim 2: MINE payoff mechanism can be used to explain the observed network structure. • Use bank employee relationships to demonstrate • Seems like the reward function for forming different relationships fits what would exist in practice Claim 3: MINE payoff mechanism can enable effective transfer to unseen environments. • Example with different network types shows how transferring and re-optimizing can lead to faster learning and one-shot example can still be useful. • Examples effectively show MINE’s ability to transfer and how transferring can be helpful. Claim 4: Network emergence game as a learned model supports meaningful strategic predictions. • Link prediction experiment shows MINE is comparable to other link prediction algorithms of interest. • Experiment effectively shows MINE can be used for strategic prediction on par with other link prediction methods, though doesn’t necessarily bring any big changes to the task of link prediction. Claim 5: A key outcome of data-driven learning of network emergence games is its ability to facilitate interpretability (discovered reward function can be analyzed to characterize the observed network properties) and generalization (the learned modules can be transferred for use beyond observed network). • Experiments seem to show this is the case.

Clarity: Generally readable, albeit quite dense.

Relation to Prior Work: A lot of good description of related work, especially threads built on. Sweeping statement about lack of prior consideration of game-theoretic equilibrium in structure learning (lines 31-32) is not accurate. See for example prior work by Honorio & Ortiz (JMLR 2015), Irfan & Ortiz (various), and Duong et al. (several papers on graphical multiagent models). Perhaps also work on learning graphical game models.

Reproducibility: No

Additional Feedback: Experimental evaluation seems to be generally well-considered and creative. For instance, the piece with swapping a leader and follower in the karate network and changing the KC value of nodes without changing their feature importance values in the network were good examples of MINE’s learning capabilities (not just learning structure, but deeper interactions). In the experiments they report fraction of links recovered, but could MINE’s learning come up with links that don’t exist in the real network? And if so, at what rate does this happen? Depending on the application, this could be important in addition to (or even more so) than just the fraction of links recovered that are accurate. Misc Notes • line 110, not sure why it is {0,1}^nx(n-1) not {0,1}^nxn • line 118, should the definition of η include “for all j in V” or something? • It seems like the paper defines the η function as the neighborhood of I (ex: line 118), but also uses N(i) to denote neighborhood (ex: line 216) • line 239, what is o with a bar on top? In section two, it defines the bar as meaning “for all agents”, but o_i appears to mean the observation for just agent i • line 256, “trajectories trajectory” • line 260, paypoff --- Post-rebuttal Our review was positive before the rebuttal and the author response did not change that.


Review 4

Summary and Contributions: Network emergence in games is often considered to as a function of the state of the network without exploiting the the sequential decision making of individual actors in the game. The authors propose a framework, MINE, that leverages the existence of Markov perfect equilibrium in multi-agent games with intuitions from IRL. The authors experiment on different datasets to evaluate on three author defined metrics quality, transferability and interpretability. The authors claim that the proposed architecture performs well.

Strengths: - The approach is novel to understand network emergence in multi-agent games. - The approach is grounded with existing literature on inverse RL. - The proposed architecture helps in learning on the toyish tasks.

Weaknesses: - The writing can be improved to better convey the results of the experiments. - The proposed method is evaluated on certain dimensions (quality, transferability, interpretability) but in the strategic link prediction task the proposed model does not perform better than the baselines. - It is not clear if the baselines fall short in the qualitative metrics (transferability, quality and interpretability), as MINE is not compared against the baselines. - The description of the proposed approach can be improved for reproducibility.

Correctness: - The intuition and theoretical intuitions of the proposed appears valid to the best of my knowledge.

Clarity: - Line 17 sentence structure - Line 246 “the the” - Line 256 -trajectory-

Relation to Prior Work: - The authors cite the relevant work and also compare against them in an experiment.

Reproducibility: No

Additional Feedback:

[Author Response · NeurIPS 2020]

We appreciate the time and efforts invested by the reviewers for examining our work and providing detailed comments that would help us in improving the overall quality of our paper. We acknowledge the general comments regarding the density of the technical content in the paper and welcome the constructive feedback on improving the clarity of the paper. We will improve the overall presentation and add more details for better accessibility. As NeurIPS allows an extra page in the final version if accepted, it would give us enough room to reorganize the material so as to alleviate the concerns of the reviewers. Having said that, we firmly believe that the current version of the paper is fairly self-contained and has all the necessary details pertinent to the significantly novel and unique contributions of our paper. We hope to convince the reviewers of the same through our clarifications below and request them to revisit their score of our work.

**R1:** Thank you for your review and clarifying questions. We respectfully disagree with your assessment that the paper needs significant expansion. The ingredients, that we leverage to propose our first of its kind contribution to learn network emergence games directly from the data, are well-established *individually* in respective game theory, graph learning and RL communities. Hence, we chose to introduce them in the main paper to expose them to the readers and expand them in Appendix A to the extent relevant to our work, but defer to the basic expertise of readers or relevant previous literature for deeper understanding of those basic concepts. **Datasets.** The use of Karate network follows standard evaluation approach from IRL literature where reward quality is tested using small example for tractability. For interpretability, we chose the bank dataset (c.f. [38] for motivation on its usefulness) as it is well known in game theory community and has been analyzed rigorously for strategic mechanisms. **Clarity.** The reward in (line 278) just represent one form of possible ground truth reward (c.f. [52] for motivation behind this form) that is used following standard evaluation procedure in IRL literature and it can take any other form with no effect on our approach. **Feedback.** Feature set description is provided in Appendix C.1. Line 215 is standard message passing architecture and we refer the reviewer to [49] for details of GNN. We use continuous actions and map output action vectors drawn from Gaussian policy to the discrete edge operations. Please refer to lines 300-325 in paper and also **R3**'s review for Figure 2(b,c). Circles in Figure 2(c) signify that Katz centrality for those agents was increased when perturbing structure for testing. You misunderstand line 299 as it exactly shows that learned objective is useful – the policy trained to optimize the learned objective, exhibit closer behavior to original network, which would not be the case if recovered objective was not useful. For transfer, the reward function remains fixed i.e. reward is not trained again on target set of agents.

**R2:** Thank you for providing your feedback in great detail that helped us to clearly understand your concerns which we address below. We wish to emphasize that, in addition to use of GNN for polices and reward function as you rightly point out, to the best of our knowledge, MINE is the first approach to formulate network emergence games as multi-agent RL problem and solve it using inverse learning framework to learn both payoffs and strategies directly from data (our most significant contribution). **Interpretability.** MINE enables discovering payoff functions from data. Simply put, interpretability analysis signifies that if a useful reward function is recovered, an agent trained to optimize such objective would receive higher reward value (utility) (during evaluation) for taking actions (achieving states) that align with real-world expectations, thereby demonstrating the agreement of learned reward with the generative mechanisms of observed data. As a case in point, for Katz centrality experiments, reward function computes utility of an agent for being in a specific structural configuration (state) at convergence. For instance, teller agents generally have low Katz centrality in real-world bank networks. So higher utility value obtained for a teller with low Katz in a given structural configuration, shows learning of useful reward function. **Evaluation.** It seems there is a misunderstanding on your part as connection order is not relevant for our evaluation. Link prediction only compares with ground truth links and qualitative experiments evaluate performance in terms of utility values, both does not need to infer connection order. **Correctness.** line 108 has gamma factor inside sum; line 162, equation continues on next line (with a multiplication between two terms); line 166, description of QRE is standard and correct but shortened. Please refer to the expansion in Appendix A and relevant papers we have cited for more details; line 186 again is a standard way to describe IRL procedure (c.f. [51]) – it aims to recover a function that rationalizes the expert behaviors with the least commitment. **Clarity.** For both the reward and policy function, we follow widely used parameter-sharing mechanism for MARL (c.f. [19]) which supports transfer. The input to these functions are agent-specific observations rather than global state (due to partial observable case) and agents are identified by their features, making transfer to different set of agents feasible. Computing reward using current state-action as input is equivalent to computing it using only next state (a standard practice in RL literature and c.f. [51] for our specific form). Please refer to our response to **R1** on **Datasets**. We provide more experimental details in Appendix C and we will address other notation issues in final version of the paper.

**R3:** Thank you for your positive reviews and accurate characterization of our contributions. For lines 31-32, we meant to say *underexplored* (typo). Thanks for interesting related works, we will add them in our discussions. For link recovery in qualitative analysis, our primary goal was to check how many strategic links are recoverable at convergence. But we agree with your overall point and we do report AUC for link prediction which would consider false positives and partially answers your question. We will also add your suggestion to qualitative analysis in final version.

**R4:** Thank you for your supportive feedback. We hope that above clarifications help to alleviate some of your related concerns. We wish to emphasize that link prediction experiments aim to show that the learned game is useful as a predictive model, a desirable but lacking property in classical game-theoretic approaches. However, note that we jointly learn network formation strategy and utility function from the data, unlike baselines that optimize task-specific hand-designed objective. Hence MINE's comparable prediction performance, along with the interpretability and transfer benefits, makes it a versatile contribution. In similar vein, we are not aware of any previous works on learning network emergence games that support transfer and interpretability dimensions, hence no baselines for qualitative analysis. The quality dimension corresponds to standard approach in IRL literature used to assess the quality of the learned reward.

[Meta-Review · NeurIPS 2020]

The reviewers felt the technical contribution is strong, connecting game theory to network analysis. However, the reviewers all felt the text was difficult to understand, and I believe their experience would be similar to much of the NeurIPS readership. Thus, I strongly encourage the authors to carefully adjust the presentation to be more friendly to researchers who are not used to the concepts from outside the usual ML concepts.